# Emotional Eating and Changes in High-Sugar Food and Drink Consumption Linked to Psychological Distress and Worries: A Cohort Study from Norway

**DOI:** 10.3390/nu15030778

**Published:** 2023-02-02

**Authors:** Elaheh Javadi Arjmand, Mitra Bemanian, Jørn Henrik Vold, Jens Christoffer Skogen, Gro Mjeldheim Sandal, Erik K. Arnesen, Silje Mæland, Lars Thore Fadnes

**Affiliations:** 1Department of Global Public Health and Primary Care, Faculty of Medicine, University of Bergen, 5020 Bergen, Norway; 2Bergen Addiction Research, Department of Addiction Medicine, Haukeland University Hospital, 5021 Bergen, Norway; 3Division of Psychiatry, Haukeland University Hospital, 5021 Bergen, Norway; 4Department of Health Promotion, Norwegian Institute of Public Health, 5808 Bergen, Norway; 5Centre for Evaluation of Public Health Measures, Norwegian Institute of Public Health, 0473 Oslo, Norway; 6Alcohol and Drug Research Western Norway, Stavanger University Hospital, 4068 Stavanger, Norway; 7Department of Psychosocial Science, University of Bergen, 5020 Bergen, Norway; 8Department of Nutrition, Institute of Basic Medical Sciences, Faculty of Medicine, University of Oslo, 0317 Oslo, Norway

**Keywords:** feeding behavior, emotional eating, psychological distress, COVID-19, Norway

## Abstract

Psychological distress is linked to unhealthy eating behaviors such as emotional eating and consumption of high-sugar food and drinks. Cross-sectional studies from early in the COVID-19 pandemic showed a high occurrence of worries and psychological distress, and this was associated with emotional eating. Few larger studies have examined how this coping pattern develops over time. This cohort study with 24,968 participants assessed changes over time in emotional eating, consumption of sugary foods as an example of unhealthy food choices, and consumption of fruits and vegetables as an example of healthy food choices. Further, associations between these and psychological distress, worries, and socio-demographic factors were assessed. Data were collected at three time points (April 2020, initially in the COVID-19 pandemic, then one and two years later). Emotional eating and intake of sugary foods and drinks were high at the start of the pandemic, followed by a reduction over time. High psychological distress was strongly associated with higher levels of emotional eating and high-sugar food intake, and lower levels of healthy eating habits. The strength of this association reduced over time. Our findings indicate the high frequency in unhealthy food choices seen early in the COVID-19 pandemic improved over time.

## 1. Introduction

A sudden change in daily routines, social distancing, and confinement may constitute stressors triggering coping mechanisms characterized by psychological or behavioral responses [1]. Stressors refer to sources of stress, and symptoms of distress are likely to occur when challenging situations are perceived to exceed coping capacities [2]. Psychological distress is a general term that refers to non-specific symptoms of depression, anxiety, and stress [3]. Studies during the early phases of the current pandemic reported psychological distress as one of the main psychosocial impacts of COVID-19 in different populations [4,5,6,7].

For human beings, food consumption is not only driven by hunger and satiety cues, but can also help to structure everyday life, create opportunities to meet people, control their physical fitness, and regulate their emotional state [8]. One major aspect of daily life that is disrupted by a pandemic and subsequent responses to curb further spread is eating habits [9]. Being in quarantine and performing social distancing reduce the access to fresh food, thus urging people to consume processed products rather than fresh produce such as fruits and vegetables [9]. On the other hand, the pandemic and its lifestyle consequences have the potential to encourage people to pursue healthy eating behaviors to ensure they are protected from the disease, as seen in previous outbreaks in Asia [10]. For example, they can have more time to prepare homemade meals instead of ready-to-eat products [11]. Considering that the pandemic is a stressful situation, the changes in eating habits may partly reflect mechanisms of coping. People may adopt maladaptive coping strategies, resulting in unhealthy eating habits. On the contrary, they could tend to adaptive coping strategies such as actively attempting to improve a situation, positive reframing, and accepting reality, all of which might reduce psychological distress and possibly act as protectors from negative eating habits [12].

The variability across individual eating behaviors can be categorized into different eating styles/food choices, one of which is emotional eating (EE). EE is eating in response to negative emotional states, such as anxiety, loneliness, or boredom [13]. Based on a model on coping with stress by Lazarus and Folkman, sensing a threat (primary appraisal), such as the recent COVID-19 pandemic, induces a range of negative emotional responses (including stress, boredom, worry, and depression). In addition, this emotional response triggers a behavioral activation or coping strategy that redirects and stimulates pleasant emotions and sensations of well-being, such as EE [1]. As part of the mental disorders group, EE is connected to the biopsychosocial model [14], along with problems related to maladaptive attitudes, cognition, and behaviors in response to adverse health events. A public health crisis, such as the COVID-19 pandemic, could negatively disrupt state aspects related to cognition and coping, and lead to a perceived state of threat [15].

Emotional eating is often linked to consuming energy-dense and sweet foods and drinks [16]. A population based study by Camilleri et al. [17], reported that with most subgroups of respondents, EE was associated with higher intakes of high-density snack foods. This was particularly notably for sweet and fatty foods, such as cakes, biscuits, pastries, chocolate, ice cream, chocolate-based products, and confectionary. Another study in Finland reported that higher EE was found to be associated with eating more sweet and non-sweet energy-dense meals, but not with eating vegetables or fruit or berries [18]. This consumption represents a way of handling negative emotions from psychological stress, and consequently, being in a stressful situation can act as a trigger to increasing the consumption of these types of food [19]. Even though some studies have assessed the EE during pandemics and other stressful life events [20,21,22,23,24], few larger studies have examined how this coping pattern develops over time and to what degree they normalize when the pandemic and its main stressors are less pronounced. 

In the Lazarus and Folkman model [1], there is a bidirectional relationship between emotions and coping, which means they can influence each other. The appraisal process after encountering a stressful situation generates emotions and, consequently, behavior change as part of a coping strategy, which modifies the individual’s relationship to the environment. This new individual–environment relationship is reappraised and could change the emotions [1]. Considering this, during the early stages of a pandemic, when people are first becoming aware of the threat and deciding how to respond to it, a threat–emotion–coping sequence is expected, and after the coping and re-evaluation of the threat have occurred, a threat–coping–emotion sequence may happen [25]. 

In this study, we assessed changes in emotional eating and food consumption in light of psychological distress and worries during the different stages of the COVID-19 pandemic. Our main objectives were to assess changes in the intake of sugary foods and drinks during different stages of the pandemic, and whether it is associated with worries and psychological distress. Correspondingly, we assessed changes in the intake of fruit and vegetables as examples of healthy foods, how prevalent EE is, and whether it is associated with worries and psychological distress. 

## 2. Materials and Methods

### 2.1. Study Characteristics; Design, Population, Data Collection and Study Sample

This is a cohort study presenting data from Bergen-in-change study [26]. A random sample of 81,170 adults from a population of 224,000 in Bergen, Western Norway, were asked to participate in a survey evaluating the effects of the COVID-19 pandemic and the non-pharmaceutical interventions adopted. The sample was representative of the general population regarding age and sex. Participants were selected by the Norwegian Digitalization Agency from a contact list. Using the web-based questionnaire platform SurveyXact, the questionnaire was distributed to the invited individuals. At the first time point, in April 2020, 29,535 (36%) participants accepted to participate in the study, and of those, 84% (*n*_0_ = 24,968) completed the required questionnaire items. Data were collected again approximately one year later, in January 2021 (*n*_1_ = 15,904), and two years later, in May 2022 (*n*_2_ = 9442). At the first time point, four to six weeks after the first wave of the pandemic (*n*_0_), several restrictions due to COVID-19 had been initiated. These measures included social distancing, the closing of educational, cultural, and training/sports/gym facilities, home-based job obligations, and the implementation of quarantine regulations. These restrictions were to some degree reduced at the second time point (*n*_1_), and substantially reduced at the last time point (*n*_2_) when there were very few restrictions/pandemic measures in place.

### 2.2. Questionnaire

The questionnaire featured questions regarding demographic data, weight, and height, and many areas of life and health in the midst of the initial COVID-19 lockdown. In the analysis for this article, the following background variables were included: age, sex, educational attainment/level, job status, household income, health and infection worries for oneself and/or family members, and worries about the consequences of COVID-19 on their employment and economic condition. Survey items related to this study, which are described in full in Appendix A, assess eating habits and EE, COVID-19-related concerns, and symptoms of psychological distress. In short, on a three-point scale ranging from not worried to substantially worried, participants were asked to rate their degree of concern over the health-related and economic repercussions of the pandemic and the lockdown. Fear of COVID-19 transmission to oneself and one’s family was among the health-related concerns. Concerns about the economy included the fear of getting laid off or seeing a decline in financial affairs. For both health-related and economy-related worries, they were considered substantial if the participants reported some levels of worries to extreme levels. EE was evaluated by asking respondents to recall the number of instances they had engaged in comfort eating or eating more in responding to feeling down or dissatisfied during the past seven days, with responses ranging from never to every day on a seven-point Likert scale [27]. In addition, participants were asked to recall how frequently, on average, they consumed high-sugar foods and drinks, as well as fruits and vegetables, over the previous 30 days. High-sugar foods were described and represented in the questionnaire as cakes, cookies, desserts, and candies, whilst high-sugar beverages contained soft drinks and soda. One serving of fruit or vegetables was defined as 80 g [28]. Psychological distress was evaluated using the 10-item version of the Hopkins symptom checklist (SCL-10) examining mental health symptoms during the previous week, with the threshold for significant psychological distress set at a mean SCL score of 1.85.

### 2.3. Study Variables, Baseline, Clinical and Sociodemographic Factors

Baseline was defined as the time of the first responses to the questionnaire. The reference level for each variable in the model were as follows: the youngest age group (18–30), living alone, living without children, lower to no levels of worries, and no psychological distress. EE was defined as a range between 0 (no symptoms last week) to 1 (every day). High-sugar food and drink intake ranged from 0 (never in the last month) to 1 (daily) in this model. The consumption of fruits and vegetables was defined as 0 (no serving/day in the last month) to 1 (equivalent of ≥10 servings/day) in this model. When presenting differences in outcome variables in Sankey plots, they were categorized accordingly: those participants who reported no EE were categorized as no EE, those who had one or two incidents of EE in last seven days were categorized as mild EE, and those who had EE more than 3 times in the last seven days were categorized as severe EE. Moreover, those participants who reported no intake of high-sugar products to 1–3 times in last month were categorized in “never/occasional” group, those who had consumed these type of products 1 to 6 times per week in the last month were considered in “moderate” group, and those who had daily intake were considered into “daily” group. Lastly, participants who reported no intake of fruit and vegetables in last month were categorized in the no fruit/vegetable group, those who had consumed one to three portions per day in the last month were considered in some fruit/vegetable group, and those who had more than four portions of fruit and vegetables daily intake were considered into plenty fruit/vegetables group.

### 2.4. Statistical Analyses

The software Stata SE 16 (StataCorp, College Station, TX, USA) was used for all the statistical analysis. The threshold for statistical significance was set to *p* < 0.05 for all analyses unless otherwise stated (sensitivity analyses for linear mixed models also presented with alpha set to *p* < 0.01). In all analyses, we defined time as years from baseline. Linear mixed model analyses were used to investigate whether the psychological distress, worries, and sociodemographic factors were associated with the three outcomes (1: emotional eating, 2: consumption of high-sugar food and drink [unhealthy foods], and 3: consumption of fruit and vegetables [healthy foods]), and to what extent they were associated with any changes in the outcome variables over time. We also performed a sensitivity analysis using ordinal logistic model for EE as an ordinal outcome variable. For most of the participants, there were not substantial significant changes in psychological distress and worries over time (Appendix A). Thus, baseline levels were used as constant predictors for the level and changes in the outcome variables. We specified the linear mixed models as a random intercept fixed slope regression model. The estimator was set to maximum likelihood. To explore whether predictors predicted changes in outcome, the interactions between these factors and time were added to the model. 

### 2.5. Ethics

The Regional Ethical Committee for Medical Research in Western Norway approved the study (REK 2020/131560). Before responding to the email survey, all participants supplied electronic informed consent; confidentiality and the right to withdraw from the study were guaranteed. The study adheres to the ethical criteria outlined in the Helsinki Declaration.

## 3. Results

In the study population, 56% were female, and 50% were under 50 years of age; 77% lived with at least one adult, 36% lived with children in their household; 65% had college- or university-level education, and 68% were fully or partly employed prior to the pandemic (Table 1 and Appendix A). Overall, 45% of the participants reported substantial health-related worries, and 19% of the participants reported substantial worries related to personal economy due to the COVID-19 pandemic. All baseline characteristics were significantly different between age groups (*p* < 0.001). There were few with large or huge changes in psychological distress over time (11 and 2%, respectively). While the symptoms of worries generally reduced over time, most changed to the neighboring categories, and few changed between the extremes (Appendix A).

### 3.1. Emotional Eating at Baseline and over Time

There was a significant decrease in EE over time (Table 2). At the baseline, EE was more common in women compared to men, but this difference decreased over time. When comparing age groups at the baseline, individuals in their 30s had more frequent EE episodes compared to adults < 30 years. In contrast, the oldest age group had less frequent episodes of EE initially, but EE slightly increased over time compared to younger age groups. High psychological distress was associated with substantially more EE at the baseline, but this association decreased over time during different stages of the pandemic. The results from ordinal logistic model were generally in similar direction. When analyzing linear mixed models with 99% confidence intervals, similar results with slightly wider confidence intervals were observed with only living with other adults at the baseline becoming non-significant, Appendix A. The changes in levels of EE over time are presented in Figure 1. At the baseline, nearly half of the participants reported no levels of EE, around one-third had mild levels of EE, and one-fifth had severe levels. Among those classified with mild EE, around half of them were later classified as having no EE and fewer developed more severe EE over time. There are some fluctuations between the one- and two-year follow-ups, but few changed between no EE and severe EE symptoms. The overall trend was that participants developed fewer symptoms of EE over time. 

### 3.2. High-Sugar Food and Drink Intake over Time

The intake of high-sugar foods and drinks increased slightly over time (Table 3). At the baseline, male participants reported a higher intake of high-sugar foods and drinks compared to females. Additionally, young adults had the highest consumption of high-sugar foods and drinks compared to all other age groups. Moreover, scoring higher on psychological distress was associated with eating and drinking more high-sugar products at the baseline. However, this relation declined over time. When analyzing linear mixed models with 99% confidence intervals, similar results with slightly wider confidence intervals were observed, and only health-related worries at the baseline became non-significant. Additionally, the time trend coefficient for psychological distress changed to non-significant, which supports the fact that the association between intake of high-sugar foods and drinks and psychological distress declined over time, Appendix A. The changes in consumption of high-sugar food and drinks over time are presented in Figure 2. At the baseline, around one fourth of the participants never or rarely consumed high-sugar products, more than half had moderate intake, and only one-tenth of them had daily consumption. Among those classified with never or occasional intake, nearly half increased their intake to moderate over time. Around two out of three of those who were classified as having a moderate intake stayed the same over time. 

### 3.3. Servings per Day of Fruits and Vegetables over Time

The consumption of fruits and vegetables did not change significantly over time (Table 4). Female participants tended to eat fruits and vegetables more frequently than males at the baseline, and this difference was stable over time. The individuals who had higher psychological distress, reported eating less fruits and vegetables at the baseline, but this association declined over time (0.022 95%CI 0.006; 0.038). When analyzing linear mixed models with 99% confidence intervals, similar results with slightly wider confidence intervals were observed and only living with other adults at the baseline changed to non-significant, Appendix A. The changes in intake of fruits and vegetables over time are presented in Figure 3. At the baseline, nearly two-thirds of the participants consumed some portions of fruits and vegetables, nearly one-third had plenty of portions in their daily intake, and only a slight number did not have any intake of fruits and vegetables. There are some fluctuations between the one- and two-year follow-ups, but the overall trend was that the intake of fruits and vegetables did not change over time.

## 4. Discussion

This study showed that emotional eating and unhealthy eating food choices were very common in the early phases of the COVID-19 pandemic, but fortunately, it substantially decreased over time with less emotional eating when the pandemic control measures had to a large degree ceased. The changes in consumption of fruits and vegetables as an example of healthy food choices did not change significantly over time; however, high psychological distress was associated with eating less fruits and vegetables initially, and this association diminished over time. Likewise, the intake of high-sugar food and drinks, as an example of unhealthy food choices, increased slightly over time, but the initial strong association between high psychological distress and intake of high-sugar food and drinks declined over time. There were few substantial changes in psychological distress over time, while regarding worries, there was a trend toward a general reduction in these symptoms, but most changed to the neighboring category, and few changed between the extremes.

A few similar studies have shown an increasing trend in EE associated with perceived psychological distress at a cross-sectional level during the COVID-19 pandemic and during previous infectious disease outbreaks, but as far as we are aware, there are few that have conducted a long term follow-up from larger populations.

Stress is thought to influence health via two distinct but interacting pathways, both through a direct, biological pathway by influencing neuroendocrine and autonomic processes, and an indirect behavioral pathway by influencing habitual and non-habitual health behaviors [30]. These pathways are likely to operate in a bi-directional protean fashion, with changes in behavior impacting biology and changes in biology influencing behavioral changes which affect health [30]. One important pathway is the hypothalamo–pituitary–adrenal axis. Acute stress activates this axis, which leads to elevated cortisol levels. Furthermore, stress and high cortisol act to control food consumption and energy expenditure. Stress is related to increased food intake, and only in a subset of the general population can it reduce food intake. Even in the case of an overall reduction in food intake, glucocorticoids are associated with consumption of foods enriched with sugary foods (or “comfort foods”). During the acute phases of stress, high cortisol levels cause negative feedback inhibition to the pituitary glands, but if the stressor is highly intense and/or prolonged, the efficacy of this feedback would markedly reduce [31]. Under this condition, glucocorticoids can have a more profound effect on high-sugar food intake and emotional eating, which due to its maladaptation to stress, acts to shut down the hypothalamo–pituitary–adrenal axis during chronic phases of stress. In this case, intraabdominal caloric storage could increase and a signal from this storage acts on the brain to reduce the chronic stress responses [32]. This could be one possible explanation of the reduction we found in sugary food and drinks intake and how emotional eating declined over time.

Another explanation for the decline in EE over time could be related to the fact that humans can modify what they perceive as stressful and how to respond to it [33]. Thus, a reduction in EE might reflect that many people adapted to the situation. We can argue that although there was a strong association between unhealthy eating habits (i.e., high consumption of high-sugar food and drink and low consumption of fruits and vegetables) and psychological distress in the earlier phases of the pandemic, this relation diminishes over time due to the fact mentioned above. 

Our previous cross-sectional study from the same sample at the first time point reported that 54% of the general population had episodes of EE in the early phases of the COVID-19 pandemic and that psychological distress was strongly associated with EE. The reduction in EE was also parallel with the reduction in the symptoms of worries over time. Additionally, worries related to personal economy and health were associated with EE [20]. A study in Switzerland examined the impact of the pandemic on comfort eating over time. The perceived consequences of the pandemic (economic, personal, and health), comfort food consumption and emotional distress were assessed at six time points. The study reported that in the first waves, the negative consequences of COVID-19 on comfort eating were strongly mediated by emotional distress, and these consequences have an indirect longitudinal effect on comfort eating through raised emotional distress [34]. 

The present study had several strengths. The large sample size means that analyses with high precision and statistical power can be performed, and yearly follow-up of the participants can provide insight into change over time during the pandemic era. Additionally, we expanded the results from the previous cross-sectional study in Norway in this longitudinally assessed study [20]. The sample is probably to some degree generalizable to other populations in high-income settings. Furthermore, our sample included both female and male participants and participants of different ages. We performed the analysis using a linear mixed model (treating EE as a continuous outcome variable) as well as a sensitivity analysis with an ordinal logistic model (treating EE as an ordinal outcome variable). Since the results from both models were generally paralleled, our analysis is likely to be robust.

One limitation of this study is that its reliance on self-report makes it susceptible to recall bias and relying on the participants’ own perceptions. However, the recall time frame was relatively short, and thus, the recall bias is likely to be less pronounced. It is also based on validated questions on psychological distress and worries [35,36], but relatively few questions regarding eating habits that have also been used in other large population-based studies, which could have enabled comparisons [37]. It could provide fewer details and nuances compared to a larger questionnaire, but larger questionnaires would have a larger risk of participants being lost to follow-up due to questionnaire fatigue. Due to a lack of evidence on emotional eating before the baseline, it is not possible to directly compare the results to pre-pandemic periods. However, EE reduced during the different stages of the pandemic. Our study also has a small inherent selection bias due to the fact that the questionnaire was in Norwegian and provided digitally, so those with limited access to the internet and limited fluency in the Norwegian language (for example, elderly inhabitants and first-generation immigrants) were less likely to participate in our study. Another limitation is that we had loss to follow-up among half of the participants over the two-year period. Still, the background factors in each of the groups were similar, and we believe that substantial selection biases influencing the observed associations are unlikely.

## 5. Conclusions

Our findings show that the early impact of the pandemic on psychological distress was associated with substantial and maladaptive eating behaviors. Fortunately, the degree of the maladaptive eating behaviors reduced over time with less emotional eating when gradually returning to a more normalized life with gradually fewer COVID-19-related restrictions. The early high frequency in unhealthy food choices and emotional eating in the early phases of the pandemic fortunately reduced considerably over time. Additionally, emotional eating and the intake of high-sugar food and drinks were initially strongly associated with psychological distress, but these associations also reduced over time. Even though the larger population improved in their initial maladaptive eating behaviors, there are still sub-groups with substantial potential for improvements.

## Figures and Tables

**Figure 1 nutrients-15-00778-f001:**
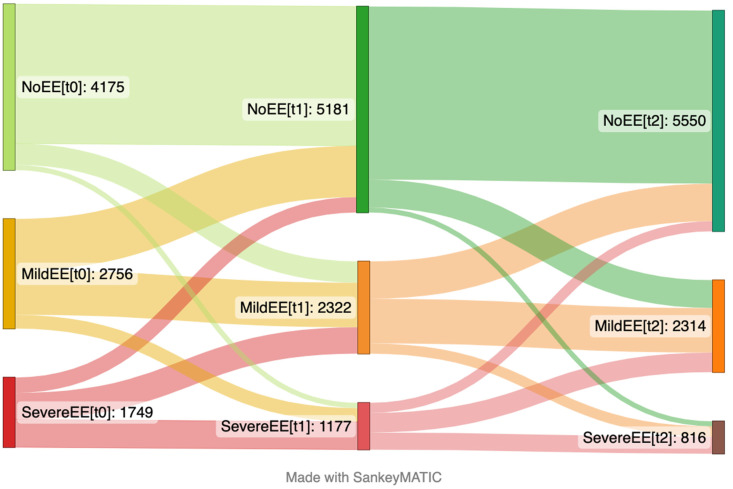
Sankey plot presenting changes in emotional eating over time from the April 2020 (**left side**) to January 2021 (**center**) and May 2022 (**right side**). The number in brackets represents time points. This plot includes the participants that answered the items focusing on emotional eating on the questionnaire at the first time point (t_0_), second time point (t_1_) and third time point (t_2_) (*n* = 8453).

**Figure 2 nutrients-15-00778-f002:**
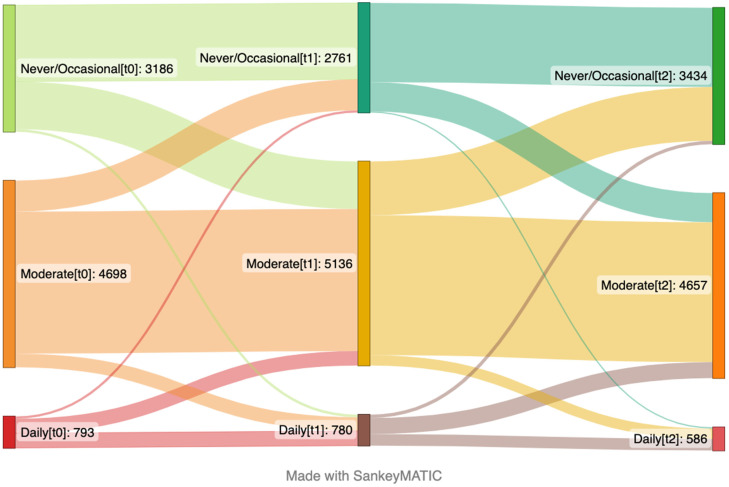
Sankey plot presenting changes in intake of high-sugar food and drinks over time from the April 2020 (**left side**) to January 2021 (**center**) and May 2022 (**right side**). The number in brackets represents time points. This plot includes the participants that answered the items focusing on intake of high-sugar food and drinks on the questionnaire at the first time point (t_0_), second time point (t_1_) and third time point (t_2_) (*n* = 8677).

**Figure 3 nutrients-15-00778-f003:**
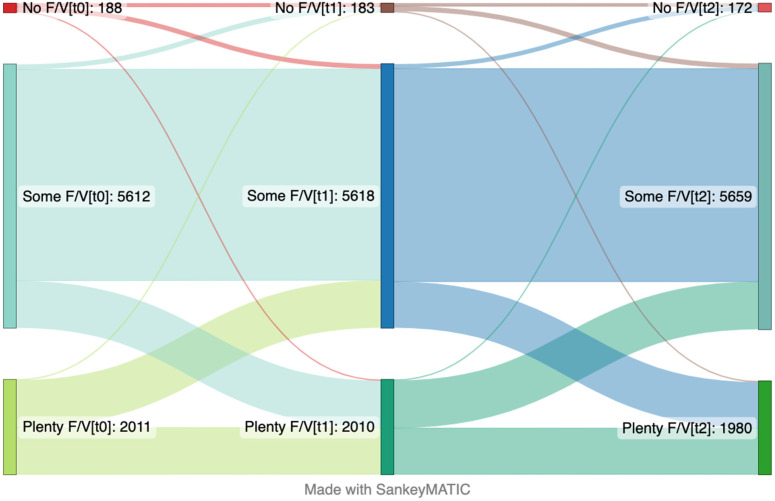
Sankey plot presenting changes in intake of fruit and vegetables (F/V) over time from the April 2020 (**left side**) to January 2021 (**center**) and May 2022 (**right side**). The number in brackets represents time points. This plot includes the participants that answered the items focusing on intake fruit and vegetables on the questionnaire at the first time point (t_0_), second time point (t_1_) and third time point (t_2_) (*n* = 7811).

**Table 1 nutrients-15-00778-t001:** Baseline characteristics of the cohort.

Age	18–30	30–40	40–50	50–60	60–70	70+
Female (%)	2200 (64)	2530 (60)	2768 (58)	2950 (56)	2133 (50)	1363 (45)
BMI categories * (%)						
Underweight	114 (4)	58 (1)	31 (1)	36 (1)	38 (1)	32 (1)
Normal	2038 (64)	1285 (55)	1500 (48)	525 (42)	43 (39)	3 (21)
Overweight	739 (23)	1278 (32)	1673 (37)	2103 (43)	1717 (43)	1154 (41)
Obese	298 (9)	516 (13)	704 (16)	808 (16)	553 (14)	285 (10)
Living with ≥1 adult (s)	2911 (85)	3415 (81)	3887 (81)	4235 (80)	3044 (72)	1821 (60)
Own children (<18y) in house	811 (24)	2595 (61)	3486 (73)	1565 (30)	334 (8)	247 (8)
Educational level						
Primary school	414 (13)	150 (4)	182 (4)	272 (5)	358 (9)	375 (13)
High school/trade school	1177 (36)	773 (19)	932 (20)	1604 (31)	1346 (32)	959 (32)
College/University	1694 (52)	3155 (77)	3510 (76)	3265 (64)	2469 (59)	1661 (55)
Employed prior to COVID-19 (%)	2178 (64)	3548 (84)	4175 (87)	4550 (86)	2415 (57)	201 (7)
Household income (%)						
Low	1041 (36)	522 (13)	462 (10)	364 (8)	234 (6)	314 (13)
Medium	1098 (38)	1944 (49)	2259 (51)	1800 (38)	1262 (35)	1204 (50)
High	734 (26)	1464 (37)	1678 (38)	2536 (54)	2109 (59)	886 (37)
Substantial worries (%)						
Economy-related worries	998 (29)	514 (22)	561 (17)	207 (16)	12 (10)	<5 (7)
Health-related worries	1889 (55)	2089 (49)	2181 (46)	2507 (48)	1559 (37)	887 (29)
High psychological distress (%)	1403 (41)	1211 (29)	935 (20)	823 (16)	475 (11)	247 (8)

* BMI, body mass index = weight (kilograms) divided by height squared (meters) (kg/m^2^). Underweight: <18.5, Normal weight: 18.5–25, Overweight: 25–30, Obese ≥ 30 [29]. The household income was modified based on the size of the family (first adult with weight 1, additional adults 0.70, and children 0.50). Norwegian krone (NOK) 250 K/year (Low), 250–500 K/year (Medium), and >500 K/year (Hight); these numbers can be converted to EUR using the exchange rate on 21 November 2022 (10.4898). Mean symptom checklist (SCL)-10 score ≥ 1.85. For worries, the health-related and economy-related worries are reported and presented separately (i.e., proportion having substantial symptoms to each of these).

**Table 2 nutrients-15-00778-t002:** Degree of emotional eating and associations to psychological distress, worries and baseline characteristics (linear mixed model presenting absolute coefficients with 0 indicating no difference/change, >0 indicating higher frequency of emotional eating within group and <0 indicating less emotional eating within group).

Estimates Presented as Coefficients (with 95% Confidence Intervals)		
	Fixed Effects	Time Trend (Per Year)
Age		
18–29	0 (reference)	0 (reference)
30–39	0.048 (0.038; 0.058)	−0.015 (−0.024; −0.006)
40–49	0.033 (0.023; 0.043)	−0.015 (−0.023; −0.006)
50–59	0.024 (0.015; 0.033)	−0.007 (−0.016; 0.001)
60–69	−0.001 (−0.011; 0.009)	−0.002 (−0.011; 0.006)
70+	−0.031 (−0.042; −0.020)	0.014 (0.004; 0.023)
Sex		
Male	0 (reference)	0 (reference)
Female	0.055 (0.050; 0.061)	−0.021 (−0.025; −0.016)
Living with other adult(s)		
No	0 (reference)	0 (reference)
Yes	−0.02 (−0.03; −0.01)	0.001 (0.000; 0.018)
Living with own children (<18 years of age)		
No	0 (reference)	0 (reference)
Yes	0.000 (−0.007; 0.006)	0.040 (0.034; 0.046)
Health-related worries		
None or some	0 (reference)	0 (reference)
Substantial	0.018 (0.013; 0.023)	−0.009 (−0.014; −0.004)
Worries related to economy		
None or some	0 (reference)	0 (reference)
Substantial	0.044 (0.037; 0.051)	−0.001 (−0.009; 0.007)
Psychological distress (0 = no to 1 = extreme)	0.63 (0.61; 0.65)	−0.18 (−0.20; −0.17)

Baseline constant of emotional eating: 0.06 (0.05; 0.07); time trend: −0.07 (−0.08; −0.06).

**Table 3 nutrients-15-00778-t003:** High-sugar food and drink intake and associations to psychological distress, worries and baseline characteristics (linear mixed model presenting absolute coefficients, with 0 indicating no difference/change, >0 indicating higher intake within group, and <0 indicating less intake within group).

Estimates Presented as Coefficients (with 95% Confidence Intervals)		
	Fixed Effects	Time Trend (Per Year)
Age		
18–29	0 (reference)	0 (reference)
30–39	−0.033 (−0.043; −0.024)	0.001 (−0.006; 0.008)
40–49	−0.062 (−0.071; −0.053)	−0.003 (−0.010; 0.003)
50–59	−0.105 (−0.114; −0.096)	−0.004 (−0.011; 0.002)
60–69	−0.139 (−0.148; −0.130)	−0.003 (−0.009; 0.004)
70+	−0.151(−0.162; −0.141)	0.004 (−0.003; 0.011)
Sex		
Male	0 (reference)	0 (reference)
Female	−0.020 (−0.025; −0.015)	−0.009 (−0.012; −0.006)
Living with other adult(s)		
No	0 (reference)	0 (reference)
Yes	0.011 (0.005; 0.016)	−0.003 (−0.010; 0.004)
Living with own children (<18 y/o)		
No	0 (reference)	0 (reference)
Yes	0.029 (0.023; 0.034)	−0.038 (−0.043; −0.033)
Health-related worries		
None or some	0 (reference)	0 (reference)
Substantial	0.004 (0.000; 0.009)	0.001 (−0.003; 0.005)
Worries related to personal economy		
None or some	0 (reference)	0 (reference)
Substantial	0.008 (0.002; 0.014)	0.004 (−0.002; 0.010)
Psychological distress (0 = no to 1 = extreme)	0.11 (0.10; 0.13)	−0.014 (−0.025; −0.003)

Baseline constant of high-sugar food and drink intake: 0.38 (0.37; 0.39), time trend: 0.034 (0.025; 0.042).

**Table 4 nutrients-15-00778-t004:** Servings per day of fruits and vegetables and associations to psychological distress, worries and baseline characteristics (linear mixed model presenting absolute coefficients with 0 indicating no difference/change, >0 indicating higher intake within group and <0 indicating less intake within group).

Effect Estimates Coefficients (with 95% Confidence Intervals)		
	Constant/Fixed Effects	Time Trend (Per Year)
Age		
18–30	0 (reference)	0 (reference)
30–40	0.006 (−0.004; 0.016)	0.007 (−0.002; 0.017)
40–50	−0.007 (−0.017; 0.003)	0.009 (−0.001; 0.019)
50–60	0.010 (0.000; 0.020)	0.006 (−0.001; 0.019)
60–70	0.030 (0.020; 0.041)	0.004 (−0.003; 0.013)
70+	0.049 (0.037; 0.060)	−0.0005 (−0.011; 0.010)
Sex		
Male	0 (reference)	0 (reference)
Female	0.045 (0.039; 0.051)	−0.002 (−0.067; 0.030)
Living with other adult(s)		
No	0 (reference)	0 (reference)
Yes	0.008 (0.001; 0.060)	−0.001 (−0.007; 0.003)
Living with own children (<18 years)		
No	0 (reference)	0 (reference)
Yes	0.001 (−0.006; 0.008)	−0.005 (−0.012; 0.002)
Health-related worries		
None or some	0 (reference)	0 (reference)
Substantial	−0.001 (−0.007; 0.005)	0.002 (−0.003; 0.008)
Worries related to personal economy		
None or some	0 (reference)	0 (reference)
Substantial	0.011 (0.004; 0.019)	−0.005 (−0.014; 0.004)
Psychological distress (0 = no to 1 = extreme)	−0.040 (−0.057; −0.022)	0.022 (0.006; 0.038)

Baseline constant of fruit and vegetables intake: 0.25 (0.24; 0.26), time trend: −0.002 (−0.015; 0.010).

## Data Availability

The data presented in this study are available on request from the corresponding author.

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
