# Peer review of "Emotional Eating and Changes in High-Sugar Food and Drink Consumption Linked to Psychological Distress and Worries: A Cohort Study from Norway"

_nutrients, 2023, doi:10.3390/nu15030778_

Round 1

Reviewer 1 Report

This study investigated emotional eating during the pandemic. In particular, the consumption of fruit, vegetables, and high-sugar food and drink relative to emotional eating. Almost 25,000 participants completed the online survey. Data were collected at 3-time points: baseline, one year later, and two years later. At each time point, emotional eating, high-sugary food and drink intake, and serving fruits and vegetables per day were the survey’s focus.

My comments focus on the results. Tables 2, 3, and 4 have a similar layout and present the fixed effects and time trend. I am having a difficult time understanding the data presented. For example, line 181 states women were “more common” to have EE than men, yet the table shows 0.55 and -0.021 for females. Since there are three-time points, I am unsure how to interpret this relationship. Also, the statistics section should be expanded to describe better what was tested.

Figure 1 is a Sankey plot and a good visual representation of the changes. There is another Sankey plot in the supplemental data. It has a better explanation of the figure. The legend of Figure 1 should be expanded.

Table 1: The vertical or horizontal percent numbers under substantial worries do not add up to 100%.

Table 1, under Substantial worries: (490 should be (49).

Figure 1: What are the numbers in the brackets? My guess is they are the time points. An explanation should be included in the figure legend.

Figure 1: For time 0, there are 8680 participants. For time 1, there are 9029 participants, and for time 2, 8640. Why does the number change?

Reviewer 2 Report

This is an interesting analysis describing changes in emotional eating and food consumption during different stages of COVID-19 pandemic. Several potential predictors of these outcomes have been considered, including psychological distress and worries related to both economy and health. Manuscript is well written and easy to follow, but some points need to be considered.

1. In the Title and in the Abstract, authors mention dietary patterns despite actually measuring consumption of three food groups. Dietary patterns refer to a broader concept, the diet as a whole. Despite authors state that these food groups serve as and indicator of dietary patterns, I would recommend to change the phrase “dietary pattern” to food consumption or food choice throughout the manuscript, to be more specific.

2. In the Abstract, line 32-33, authors conclude “Our findings indicate that the initial increase in unhealthy eating patterns in response to the COVID-19 pandemic improved over time.” However, the data is collected at three time points, out of which the first was, when the pandemic had already started (April 2020). Hence, there is no data to support the claim that there was an “initial increase in response to pandemic”. Here and throughout the manuscript authors need to be very careful not to make causal claims between pandemic and values measured at the first measurement point.

3. There is something wrong with the references, for example ref nro 17 mentioned in line 72 is Konttinen et al., not Geraldine et al. And ref nro 18 is not based on data from Finland. I suggest authors to check all the references.

4. In the Methods section: Most of the variables are categorized somehow, but the cut-points for categorization are not presented. These need to be included in the Methods section. For example, how “substantially worried” (economic/health related) or mild/severe EE is defined?

5. In Tables 2-4 there seems to be (beta?) coefficients for outcomes (first lines). This is confusing, is it a mistake?

6. In lines 224-225 authors conclude “more healthy eating patterns when the pandemic control measures had to large degree ceased”. I’m wondering whether the results support this claim, because results show no change in intake of vegetables over time.

7. Since the main focus of the study is the change in EE and consumption of sugary products and vegetables (measured as continuous variables), it would be useful to see a graphical presentation of these changes over time. Now it is a bit hard to grasp an idea of the magnitude of these changes.

8. It is an interesting fact that psychological stress and worries related to health and economics did not change over time (mentioned in lines 152-153). This could be discussed a bit more, because the whole analysis is based on a theory that COVID-19 is a major stressor and potentially affects EE and dietary intake. But was it so after all?

9. In line 257-258 there is a reference missing for “one study in Norway”. Was it conducted in same sample? If so, it should be mentioned. Likewise, there is a reference missing to “previous study” in line 270.

10. The questions used to measure the variables of interest are very simple, which is nicely stated in the Limitations section of the Discussion. The authors mention in lines 276-277 that the questions have been validated, but there is no information about this validation. It needs to be added in the Methods section.

Reviewer 3 Report

I would like to thank the authors and the Editorial Board for the opportunity to review the article submitted to Nutrients. The authors' manuscript refers to a very important topic, the relationship between psychological distress and dietary behaviour. The presented manuscript shows the results based on a very large sample of Norwegian citizens. I believe that it reflects very important results and should be published. As with any research project, the presented article is not free of limitations. Below I suggest some minor changes that might improve the overall high quality of the authors’ manuscript. I believe that the author’s manuscript would benefit from the following changes:

·         Theoretical background behind the authors’ study should be strengthened. In the introduction, the authors refer to various research data and one theoretical model. Authors refer to Lazarus & Folkman’s transactional stress model only vaguely. I highly recommend that the authors describe the coping with stress mechanism more deeply with the reference to emotional eating.

·         Table 1 frequencies could be analysed with the usage of Pearson’s chi-square test with a proper effect size measure (Cramer’s V or Yule’s Phi).

·         All used coefficients could be described in more detail. For example, authors write “with substantially more EE at baseline (0.63, Cl 0.61; 186 0.65)”. Since the authors performed a cohort study, I assume that the value of 0.63 represents RR (relative risk ratio coefficient). It might be unknown to a reader (which can be new to the field) if authors refer to OR or RR values.

·         The presented confidence intervals are very “tight”. It is due to the very large sample size (CI and alpha are highly related to the number of study participants; see Lin et al., 2013; Faber & Fonseca, 2014; Lakens, 2022). Additionally to the presented results, I suggest that the authors re-analyse their data with the use of alpha=0.01 instead of alpha=0.05 as their threshold value and/or analyse their data on a smaller sample pulled out of the existing data (with resampling methods, such as bootstrapping etc).
